# VeRA: Vector-based Random Matrix Adaptation

**Dawid J. Kopiczko**[*†]
QUVA Lab
University of Amsterdam

**Tijmen Blankevoort**
Qualcomm AI Research[‡]

**Yuki M. Asano**
QUVA Lab
University of Amsterdam

## Abstract

Low-rank adapation (LoRA) is a popular method that reduces the number of trainable parameters when finetuning large language models, but still faces acute storage challenges when scaling to even larger models or deploying numerous per-user or per-task adapted models. In this work, we present **Ve**ctor-based **R**andom Matrix **A**daptation (VeRA)[1], which significantly reduces the number of trainable parameters compared to LoRA, yet maintains the same performance. It achieves this by using a single pair of low-rank matrices shared across all layers and learning small scaling vectors instead. We demonstrate its effectiveness on the GLUE and E2E benchmarks, image classification tasks, and show its application in instruction-tuning of 7B and 13B language models.

## 1 Introduction

In the era of increasingly large and complex language models, the challenge of efficient adaptation for specific tasks has become more important than ever. While these models provide powerful capabilities, their extensive memory requirements pose a significant bottleneck, particularly when adapting them for personalized use. Consider, for example, a cloud-based operating system assistant that continuously learns from and adapts to individual user behaviors and feedback. The need to store multiple checkpoints of finetuned models for each user rapidly escalates the required storage, even more so when multiple tasks come into play.

The situation is further exacerbated when we look at the state-of-the-art models like GPT-4 (OpenAI, 2023). Finetuning techniques like LoRA (Hu et al., 2022), while effective, still introduce considerable memory overhead. As an illustrative example, applying LoRA with a rank of 16 to the query and value layers of GPT-3 (Brown et al., 2020) would demand at least 288MB of memory, if stored in singe-precision – at a million finetuned weights, e.g., one per user, that would amount to 275TB.

Given the recent proliferation of language models and their deployment in personalized assistants, edge devices, and similar applications, efficient adaptation methods are paramount. We believe there is untapped potential for even more efficient approaches. Previous work (Aghajanyan et al., 2021) pointed out the low intrinsic dimensionality of pretrained models' features. These studies reported numbers much lower than the trainable parameters used in LoRA, suggesting there is room for improvement.

In parallel to this, recent research has shown the surprising effectiveness of models utilizing random weights and projections (Peng et al., 2021; Ramanujan et al., 2020; Lu et al., 2022; Schrimpf et al., 2021; Frankle et al., 2021). Such models serve as the basis of our proposed solution, **Ve**ctor-based **R**andom Matrix **A**daptation (VeRA), which minimizes the number of trainable parameters introduced during finetuning by reparametrizing the weights matrices. Specifically, we employ "scaling vectors" to adapt a pair of frozen random matrices shared between layers. With this approach, many more versions of the model can reside in the limited memory of a single GPU.

---

[*]dj.kopiczko@gmail.com

[†]Datasets were solely downloaded and evaluated by the University of Amsterdam.

[‡]Qualcomm AI Research is an initiative of Qualcomm Technologies, Inc.

[1]Website: https://dkopi.github.io/vera/

In summary, our main contributions are as follows:

- We introduce a novel finetuning method with no additional inference time cost. Our method further reduces the number of trainable parameters compared to the state-of-the-art LoRA method, while yielding comparable results.

- We compare our approach with LoRA and other parameter-efficient adaptation methods on the natural language understanding (GLUE) and natural language generation (E2E) benchmarks, and compare against LoRA on instruction-following and image classification tasks.

- We perform an ablation study to better understand the individual components of our method and their effects on performance.

## 2 RELATED WORK

**Low-Rank Adaptation (LoRA).** LoRA offers an innovative solution to the computational challenges posed by the finetuning of large pretrained language models. Introduced by Hu et al. (2022), the method employs low-rank matrices to approximate the weight changes during finetuning, effectively reducing the number of parameters that need to be trained. Among its advantages, LoRA significantly lowers the hardware barrier for finetuning by reducing the need for gradient calculation and optimizer state maintenance for most parameters. It can also work with quantized model weights (Dettmers et al., 2023), reducing the requirements even further. Furthermore, LoRA modules are easily swappable, making task-switching efficient and less resource-intensive. Importantly, and different to adapter-based finetuning approaches (Houlsby et al., 2019; Lin et al., 2020; Pfeiffer et al., 2021; Rücklé et al., 2021), LoRA incurs no additional inference time cost when deployed, as the trainable matrices can be merged with the frozen weights.

Based on this, AdaLoRA (Zhang et al., 2023b) extends the LoRA method, introducing dynamic rank adjustment for the low-rank matrices during finetuning. The core idea is to optimally distribute the parameter budget by selectively pruning less important components of the matrices based on an importance metric.

**Parameter Efficiency in Existing Methods** While methods such as LoRA have shown significant improvements in finetuning performance, they still require a considerable amount of trainable parameters. According to Aghajanyan et al. (2021), the upper bound for *intrinsic dimensions* is much smaller than what is typically utilized in such methods. For instance, the $d_{90}{}^2$ for RoBERTa$_{\text{base}}$ is reported to be 896, whereas authors of the LoRA paper reported using 0.3M trainable parameters for this model, suggesting that the parameter count could be reduced further.

Although AdaLoRA takes steps in this direction by dynamically allocating parameters to more critical layers, we posit that a different approach could achieve substantial parameter reduction, while tolerating a marginal performance degradation. This sets the stage for the method we introduce in the following section.

**Random Models and Projections.** The concept of using random matrices and projections for model efficiency is supported by multiple strands of research. Frankle & Carbin (2019) identified that randomly-initialized neural networks contain subnetworks that are capable of reaching high performance when trained. Meanwhile, Ramanujan et al. (2020) revealed that there exist subnetworks that can achieve impressive results even in the absence of training. Aghajanyan et al. (2021) showed that training only a small number of parameters, randomly projected back into the full space, could achieve 90% of the full-parameter model performance. Ruiz et al. (2023) introduced a parameter-efficient finetuning method for personalization of text-to-image models, utilising random frozen matrices inside LoRA. Other works (Lu et al., 2022; Schrimpf et al., 2021; Frankle et al., 2021) have shown that frozen, randomly initialized models, with small sections finetuned, can perform surprisingly well.

---

[2]The smallest dimension $d$ that provides a *satisfactory solution*, which is 90% of the full training metric, as defined by Li et al. (2018).

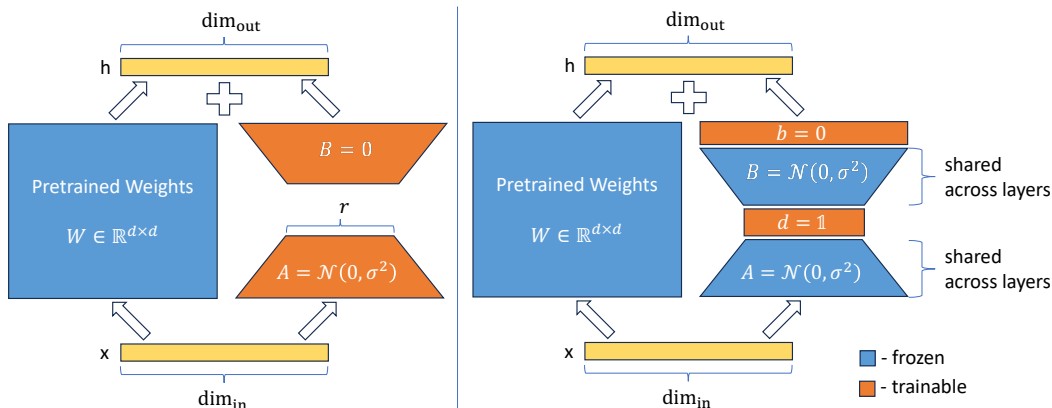

Figure 1: Schematic comparison of LoRA (left) and VeRA (right). LoRA updates the weights matrix $W$ by training the low-rank matrices $A$ and $B$, with intermediate rank $r$. In VeRA these matrices are frozen, shared across all layers, and adapted with trainable vectors $d$ and $b$, substantially reducing the number of trainable parameters. In both cases, low-rank matrices and vectors can be merged into original weights matrix $W$, introducing no additional latency.

Collectively, these works create a compelling case for the utilization of frozen random matrices in finetuning methods, providing both a theoretical and an empirical foundation for the approach taken in this paper.

## 3 METHOD

In this section, we introduce Vector-based Random Matrix Adaptation, a novel parameter-efficient finetuning method that builds upon and extends the state-of-the-art method, LoRA. The central innovation in VeRA lies in the reparameterization of the low-rank matrices. Specifically, we freeze a single pair of randomly initialized matrices, shared across all adapted layers, and introduce trainable scaling vectors that allow for layer-wise adaptation, as shown in Figure 1. Similarly to LoRA, trained scaling vectors along with low-rank matrices can be merged into original weights, eliminating additional inference latency.

### 3.1 METHOD FORMULATION

LoRA (Hu et al., 2022) finetunes a matrix product of two low-rank matrices to adapt large-language models for a new task. Formally, for a pretrained weight matrix $W_0 \in \mathbb{R}^{m \times n}$, the weight update $\Delta W$ is constrained to a low-rank decomposition, as expressed in Equation 1

$$h = W_0 x + \Delta W x = W_0 x + \underline{B}\,\underline{A} x, \tag{1}$$

where we undeline the parameters updated via gradient descent. This approximation enables the model to keep the original weight $W_0$ frozen while optimizing only the new low-rank matrices $A$ and $B$. These matrices are much smaller in size than the original matrix due to their rank-reduced nature. $A$ has shape $m \times r$ and $B$ has shape $r \times n$, where $r \ll \min(m, n)$ serves as the bottleneck dimension. In contrast, our VeRA method is expressed as:

$$h = W_0 x + \Delta W x = W_0 x + \underline{\Lambda_b} B \underline{\Lambda_d} A x \tag{2}$$

In this approach, $B$ and $A$ are *frozen, random*, and *shared across layers*, while the scaling vectors $b$ and $d$ are *trainable*, and formally denoted by diagonal matrices $\Lambda_b$ and $\Lambda_d$. This approach can effectively scale and disable rows and columns of both $A$ and $B$, allowing for layer-wise adaptation with a minimal number of trainable parameters. Note that in this setup, $B \in \mathbb{R}^{m \times r}$ and $A \in \mathbb{R}^{r \times n}$

are not required to be low-rank. This is because they remain static and we do not need to store their values. Instead, varying $r$ leads to a linear increase in the number of trainable parameters via $d \in \mathbb{R}^{1 \times r}$.

## 3.2 PARAMETER COUNT

Table 1: Theoretical memory required to store trained VeRA and LoRA weights for RoBERTa$_{\text{base}}$, RoBERTa$_{\text{large}}$ and GPT-3 models. We assume that LoRA and VeRA methods are applied on query and key layers of each transformer block.

|  | Rank | LoRA | | VeRA | |
| --- | --- | --- | --- | --- | --- |
|  |  | # Trainable Parameters | Required Bytes | # Trainable Parameters | Required Bytes |
| BASE | 1 | 36.8K | 144KB | 18.4K | 72KB |
|  | 16 | 589.8K | 2MB | 18.8K | 74KB |
|  | 256 | 9437.1K | 36MB | 24.5K | 96KB |
| LARGE | 1 | 98.3K | 384KB | 49.2K | 192KB |
|  | 16 | 1572.8K | 6MB | 49.5K | 195KB |
|  | 256 | 25165.8K | 96MB | 61.4K | 240KB |
| GPT-3 | 1 | 4.7M | 18MB | 2.4M | 9.1MB |
|  | 16 | 75.5M | 288MB | 2.8M | 10.5MB |
|  | 256 | 1207.9M | 4.6GB | 8.7M | 33MB |

We use $L_{\text{tuned}}$ to denote the number of finetuned layers and $d_{\text{model}}$ to represent the dimension of these layers. The number of trainable parameters in VeRA is then governed by $|\Theta| = L_{\text{tuned}} \times (d_{\text{model}} + r)$, contrasting with LoRA's $|\Theta| = 2 \times L_{\text{tuned}} \times d_{\text{model}} \times r$. Specifically, for the lowest rank (i.e., $r = 1$), VeRA requires approximately half the trainable parameters of LoRA. Moreover, as the rank increases, VeRA's parameter count increases by $L_{\text{tuned}}$ for each increment, a substantial saving compared to LoRA's $2L_{\text{tuned}}d_{\text{model}}$. This parameter efficiency becomes notably significant in the context of extremely deep and wide models, such as GPT-3 (Brown et al., 2020), which has 96 attention layers and a hidden size of 12288.

Building on this efficiency, the main advantage of VeRA is its minimal memory footprint for storing the trained weight adjustments. Because the random frozen matrices can be regenerated from a random number generator (RNG) seed, these do not need to be stored in memory. This substantially reduces the memory requirement, which is now limited to the bytes needed for the trained $b$ and $d$ vectors and a single RNG seed. The memory efficiency in comparison to LoRA is shown in Table 1.

## 3.3 INITIALIZATION STRATEGIES

- **Shared Matrices**: In our method, we employ Kaiming initialization (He et al., 2015) for the frozen low-rank matrices $A$ and $B$. By scaling the values based on matrix dimensions, it ensures that a matrix product of $A$ and $B$ maintains a consistent variance for all ranks, eliminating the need to finetune the learning rate for each rank.

- **Scaling Vectors**: The scaling vector $b$ is initialized to zeros, which aligns with the initialization of matrix $B$ in LoRA and ensures that the weight matrix is unaffected during the first forward pass. The scaling vector $d$ is initialized with a single non-zero value across all its elements, thereby introducing a new hyperparameter that may be tuned for better performance.

Figure 1 illustrates example initializations for the low-rank matrices and scaling vectors in VeRA. Specifically, the low-rank matrices are initialized using a normal distribution, and the $d$ vector is initialized with ones. Note that alternative initializations, such as uniform distribution for $A$ and $B$, and other non-zero constants for $d$, are also explored in our experiments.

## 4 EXPERIMENTS

In this section, we conduct a series of experiments to evaluate our finetuning method. We start by comparing our approach to LoRA and other baselines on the GLUE and E2E benchmarks. Following

this, we turn our attention to instruction-tuning of Llama models, and image classification with Vision Transformers. Next, we select one task and vary the rank for both methods, LoRA and VeRA, to examine how performance scales with the number of trainable parameters. Lastly, an ablation study sheds light on the importance of each component in our method, including the influence of different initializations.

**Baselines.** We compare VeRA to the following baselines:

- *Full finetuning* - the model is initialized with pretrained weights and all parameters are being trained.
- *Bitfit* - this baseline involves the sole finetuning of bias vectors, keeping all other parameters fixed. This technique has been investigated in depth by Zaken et al. (2022).
- *Adapter tuning* - initially introduced by Houlsby et al. (2019), involves the integration of adapter layers between the self-attention and MLP modules, followed by a residual connection. This setup includes two fully connected layers and a nonlinearity and is denoted as **Adapter$^H$**. A variation by Lin et al. (2020), **Adapter$^L$**, employs the adapter layer solely after the MLP module and subsequent to a LayerNorm. This closely resembles an alternative design suggested by Pfeiffer et al. (2021), referred to as **Adapter$^P$**. Another baseline, termed AdapterDrop by Rücklé et al. (2021), enhances efficiency by omitting certain adapter layers and is represented as **Adapter$^D$**.
- *LoRA* (Hu et al., 2022) - as introduced in the earlier section.

## 4.1 GLUE BENCHMARK

We evaluate our approach on the General Language Understanding Evaluation (GLUE) benchmark (Wang et al., 2019), employing the RoBERTa$_{base}$ and RoBERTa$_{large}$ models (Liu et al., 2019). For RoBERTa$_{base}$ we use a rank of 1024, and for RoBERTa$_{large}$ a rank of 256. The shared matrices are initialized using the uniform version of Kaiming initialization as implemented in PyTorch (Paszke et al., 2019), with an initial value of 0.1 for the $d$ vector.

Our experimental setup generally aligns with that of Hu et al. (2022), applying our method to the query and value projection matrices in each self-attention module and fully training the classification head. Unlike Hu et al. (2022), who used an additional hyperparameter $\alpha$ to adjust gradients for the adapted layers, we introduce separate learning rates for the classification head and the adapted layers. We determine the learning rates and the number of training epochs through hyperparameter tuning; for detailed settings, refer to the Table 8 in Appendix A. The batch size is set to 64 for RoBERTa$_{base}$ and 32 for RoBERTa$_{large}$, with maximum sequence lengths of 512 and 128 respectively.

Due to time constraints and budget limitations, we omit the time-intensive MNLI and QQP tasks, thus forgoing the use of the *MNLI trick*[3] for tasks MRPC, RTE, and STS-B. In line with Hu et al. (2022), we report the number of trainable parameters attributable to the finetuned layers, explicitly excluding the classification head, which is trained in a standard way. We perform 5 runs with different random seeds, recording the best epoch's outcome for each run, and report the median of these results.

**Results.** Table 2 reveals that VeRA performs competitively with LoRA across both models, yet achieves these results with an order of magnitude fewer parameters.

## 4.2 E2E BENCHMARK

For the E2E benchmark (Novikova et al., 2017), we follow the experimental setup from Hu et al. (2022) and finetune the GPT-2 (Radford et al., 2019) Medium and Large models. For LoRA we use the implementation and set of hyperparameters provided in Hu et al. (2022), while for VeRA we change the rank and learning rate, both of which are tuned. Table with all hyperparameters used can be found in Appendix A.

---

[3]For the RoBERTa$_{base}$ model and MRPC, RTE and STS-B tasks, Hu et al. (2022) initialized the model with the best weights finetuned on the MNLI task.

Table 2: Results for different adaptation methods on the GLUE benchmark. We report Matthew's correlation for CoLA, Pearson correlation for STS-B, and accuracy for the remaining tasks. In all cases, higher values indicate better performance. Results of all methods except VeRA are sourced from prior work (Hu et al., 2022; Zhang et al., 2023a). VeRA performs on par with LoRA with an order of magnitude fewer parameters.

| | Method | # Trainable Parameters | SST-2 | MRPC | CoLA | QNLI | RTE | STS-B | Avg. |
|---|---|---|---|---|---|---|---|---|---|
| BASE | FT | 125M | 94.8 | 90.2 | 63.6 | 92.8 | 78.7 | 91.2 | 85.2 |
| | BitFit | 0.1M | 93.7 | **92.7** | 62.0 | 91.8 | 81.5 | 90.8 | 85.4 |
| | Adpt$^D$ | 0.3M | $94.2_{\pm0.1}$ | $88.5_{\pm1.1}$ | $60.8_{\pm0.4}$ | $93.1_{\pm0.1}$ | $71.5_{\pm2.7}$ | $89.7_{\pm0.3}$ | 83.0 |
| | Adpt$^D$ | 0.9M | $94.7_{\pm0.3}$ | $88.4_{\pm0.1}$ | $62.6_{\pm0.9}$ | $93.0_{\pm0.2}$ | $75.9_{\pm2.2}$ | $90.3_{\pm0.1}$ | 84.2 |
| | LoRA | 0.3M | $\mathbf{95.1}_{\pm0.2}$ | $89.7_{\pm0.7}$ | $63.4_{\pm1.2}$ | $\mathbf{93.3}_{\pm0.3}$ | $\mathbf{86.6}_{\pm0.7}$ | $\mathbf{91.5}_{\pm0.2}$ | **86.6** |
| | VeRA | **0.043M** | $94.6_{\pm0.1}$ | $89.5_{\pm0.5}$ | $\mathbf{65.6}_{\pm0.8}$ | $91.8_{\pm0.2}$ | $78.7_{\pm0.7}$ | $90.7_{\pm0.2}$ | 85.2 |
| LARGE | Adpt$^P$ | 3M | $96.1_{\pm0.3}$ | $90.2_{\pm0.7}$ | $\mathbf{68.3}_{\pm1.0}$ | $\mathbf{94.8}_{\pm0.2}$ | $83.8_{\pm2.9}$ | $92.1_{\pm0.7}$ | 87.6 |
| | Adpt$^P$ | 0.8M | $\mathbf{96.6}_{\pm0.2}$ | $89.7_{\pm1.2}$ | $67.8_{\pm2.5}$ | $\mathbf{94.8}_{\pm0.3}$ | $80.1_{\pm2.9}$ | $91.9_{\pm0.4}$ | 86.8 |
| | Adpt$^H$ | 6M | $96.2_{\pm0.3}$ | $88.7_{\pm2.9}$ | $66.5_{\pm4.4}$ | $94.7_{\pm0.2}$ | $83.4_{\pm1.1}$ | $91.0_{\pm1.7}$ | 86.8 |
| | Adpt$^H$ | 0.8M | $96.3_{\pm0.5}$ | $87.7_{\pm1.7}$ | $66.3_{\pm2.0}$ | $94.7_{\pm0.2}$ | $72.9_{\pm2.9}$ | $91.5_{\pm0.5}$ | 84.9 |
| | LoRA-FA | 3.7M | 96.0 | 90.0 | 68.0 | 94.4 | 86.1 | 92.0 | 87.7 |
| | LoRA | 0.8M | $96.2_{\pm0.5}$ | $90.2_{\pm1.0}$ | $68.2_{\pm1.9}$ | $\mathbf{94.8}_{\pm0.3}$ | $85.2_{\pm1.1}$ | $\mathbf{92.3}_{\pm0.5}$ | **87.8** |
| | VeRA | **0.061M** | $96.1_{\pm0.1}$ | $\mathbf{90.9}_{\pm0.7}$ | $68.0_{\pm0.8}$ | $94.4_{\pm0.2}$ | $\mathbf{85.9}_{\pm0.7}$ | $91.7_{\pm0.8}$ | **87.8** |

Table 3: Results for different adaptation methods on the E2E benchmark and GPT2 Medium and Large models. Results with ([1,2,3]) are taken from prior work: [1](Hu et al., 2022), [2](Valipour et al., 2022), [3](Zi et al., 2023). VeRA outperforms LoRA with 3 and 4 times less trainable parameters, for GPT2 Medium and Large respectively.

| | Method | # Trainable Parameters | BLEU | NIST | METEOR | ROUGE-L | CIDEr |
|---|---|---|---|---|---|---|---|
| MEDIUM | FT[1] | 354.92M | 68.2 | 8.62 | 46.2 | 71.0 | 2.47 |
| | Adpt$^L$[1] | 0.37M | 66.3 | 8.41 | 45.0 | 69.8 | 2.40 |
| | Adpt$^L$[1] | 11.09M | 68.9 | 8.71 | 46.1 | 71.3 | 2.47 |
| | Adpt$^H$[1] | 11.09M | 67.3 | 8.50 | 46.0 | 70.7 | 2.44 |
| | DyLoRA[2] | 0.39M | 69.2 | 8.75 | 46.3 | 70.8 | 2.46 |
| | AdaLoRA[3] | 0.38M | 68.2 | 8.58 | 44.1 | 70.7 | 2.35 |
| | LoRA | 0.35M | 68.9 | 8.69 | 46.4 | 71.3 | **2.51** |
| | VeRA | **0.098M** | **70.1** | **8.81** | **46.6** | **71.5** | 2.50 |
| LARGE | FT[1] | 774.03M | 68.5 | 8.78 | 46.0 | 69.9 | 2.45 |
| | Adpt$^L$[1] | 0.88M | 69.1 | 8.68 | 46.3 | 71.4 | 2.49 |
| | Adpt$^L$[1] | 23.00M | 68.9 | 8.70 | 46.1 | 71.3 | 2.45 |
| | LoRA | 0.77M | 70.1 | 8.80 | 46.7 | **71.9** | 2.52 |
| | VeRA | **0.17M** | **70.3** | **8.85** | **46.9** | 71.6 | **2.54** |

**Results.** We report results from the last epoch. Table 3 shows that VeRA outperforms LoRA with 3 and 4 times less trainable parameters, for GPT2 Medium and Large respectively.

## 4.3 INSTRUCTION TUNING

Instruction tuning is a process by which language models are finetuned to follow specific instructions more effectively (Ouyang et al., 2022). We demonstrate the efficacy of VeRA in enabling Llama (Touvron et al., 2023a) and Llama2 (Touvron et al., 2023b) models to follow instructions using only 1.6M and 2.4M trainable parameters, for 7B and 13B variants respectively, in contrast to 159.9M and 250.3M trainable parameters when employing LoRA with a rank of 64 as proposed by Dettmers et al. (2023).

We perform finetuning using both LoRA and VeRA, by applying both methods on all linear layers except the top one, similarly to Dettmers et al. (2023). Additionally, we leverage the quantization techniques from Dettmers et al. (2023) to train the model on a single GPU.

For our experiment, we employ the Alpaca dataset (Taori et al., 2023), specifically its cleaned version[4]. This dataset comprises 51K instructions and demonstrations and is suitable for instruction-tuning. The cleaned version corrects multiple issues such as hallucinations, merged instructions, and empty outputs. We train for one epoch, preceded by a learning rate sweep.

We evaluate finetuned models on MT-Bench (Zheng et al., 2023), by generating model responses to a pre-defined set of 80 multi-turn questions and subsequently evaluating these using GPT-4 (OpenAI, 2023). GPT-4 reviews the answers and assigns a quantitative score on a scale of 10 to each response. We present the average scores alongside the number of trainable parameters in Table 4.

Table 4: Average scores on MT-Bench assigned by GPT-4 to the answers generated by models fine-tuned with VeRA and LoRA methods, and the base Llama 13B model. VeRA closely matches performance of LoRA on the instruction-following task, with 100x reduction in trainable parameters.

| Model | Method | # Parameters | Score |
|---|---|---|---|
| Llama 13B | - | - | 2.61 |
| LLAMA 7B | LoRA | 159.9M | 5.03 |
| | VeRA | 1.6M | 4.77 |
| LLAMA 13B | LoRA | 250.3M | 5.31 |
| | VeRA | 2.4M | 5.22 |
| LLAMA2 7B | LoRA | 159.9M | 5.19 |
| | VeRA | 1.6M | 5.08 |
| LLAMA2 13B | LoRA | 250.3M | 5.77 |
| | VeRA | 2.4M | 5.93 |

We find that despite the 100x reduction in the number of trainable parameters, our method closely matches the performance of LoRA-based finetuning.

## 4.4 IMAGE CLASSIFICATION

To evaluate the method on the image classification task, we adapt Vision Transformer (ViT) (Dosovitskiy et al., 2021), Base and Large variants, on datasets - CIFAR100 (Krizhevsky, 2009), Food101 (Bossard et al., 2014), Flowers102 (Nilsback & Zisserman, 2008), and RESISC45 (Cheng et al., 2017). For each dataset we train on a subset of 10 samples per class, and evaluate on the full test set (CIFAR100, Food101, Flowers102) or on all the remaining samples (RESISC45). We use weights of ViT models pretrained on the ImageNet-21k (Deng et al., 2009) dataset.

We evaluated LoRA and VeRA methods applied on the query and value layers of ViT, along with two baselines - fully-finetuned model (referred to as *Full*), and training the classification head only (referred to as *Head*). Similarly to the GLUE benchmark, we use rank 8 for LoRA, and rank 256 for VeRA. We tuned learning rates for all methods and reported results after 10 epochs in Table 5. The reported parameter count excludes the classification head, which has to be trained in all methods.

We find that VeRA approaches performance of LoRA on the Base model for three datasets and outperforms it for Flowers102, despite using over 10x fewer trainable parameters. For ViT-Large, it outperforms LoRA for three datasets: CIFAR100, Flowers102 and RESISC45.

## 4.5 SCALING THE NUMBER OF TRAINABLE PARAMETERS

Finally, we investigate the trade-offs involved in parameter scalability for both LoRA and our method using the RoBERTa$_{large}$ model on the RTE task from the GLUE benchmark. We use a set of ranks $r = \{1, 4, 16, 64, 256, 1024\}$ for VeRA and $r = \{1, 2, 4, 8, 16, 32, 64\}$ for LoRA, and observe the trade-off between trainable parameters and the accuracy. We replicate each configuration five times for different random seeds, and report the median of results. For LoRA, we employ the HuggingFace PEFT (Mangrulkar et al., 2022) implementation, adhering to the hyperparameters specified in Hu et al. (2022). Our own method uses the same hyperparameters as employed in the

---

[4]https://huggingface.co/datasets/yahma/alpaca-cleaned

Table 5: Vision models finetuned with VeRA and LoRA on different image classification datasets. VeRA approaches performance of LoRA for the smaller model, and outperforms it in the case of the large model, with over 10x fewer trainable parameters.

| | Method | # Trainable Parameters | CIFAR100 | Food101 | Flowers102 | RESISC45 |
|---|---|---|---|---|---|---|
| ViT-B | Head | - | 77.7 | 86.1 | 98.4 | 67.2 |
| | Full | 85.8M | **86.5** | **90.8** | 98.9 | **78.9** |
| | LoRA | 294.9K | 85.9 | 89.9 | 98.8 | 77.7 |
| | VeRA | **24.6K** | 84.8 | 89.0 | **99.0** | 77.0 |
| ViT-L | Head | - | 79.4 | 76.5 | 98.9 | 67.8 |
| | Full | 303.3M | 86.8 | 78.7 | 98.8 | **79.0** |
| | LoRA | 786.4K | 87.0 | **79.5** | 99.1 | 78.3 |
| | VeRA | **61.4K** | **87.5** | 79.2 | **99.2** | 78.6 |

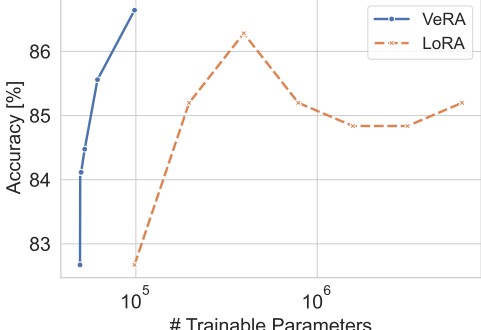

Figure 2: Performance of LoRA and VeRA methods for varying ranks on RTE task.

Figure 3: Magnitude of the adapted $d$ vector for query and value matrices across layers for RoBERTa-L on the RTE task.

RTE experiments from the previous subsection. The results, depicted in Figure 2, reveal that our method is significantly more parameter-efficient. Notably, when the higher-rank VeRA has the same number of parameters as standard LoRA, it outperforms LoRA by 4 accuracy percentage points.

## 4.6 ABLATION STUDY

In this section, we conduct an ablation study to examine the impact of individual components of our method. All subsequent experiments focus on the MRPC and RTE tasks and utilize the RoBERTa$_{\text{large}}$ model. We adhere to the hyperparameters used in previous experiments, modifying only the component under investigation for each test. Each experiment is run with 5 random seeds, and we report the mean and standard deviation of the results.

**Single Scaling Vector** We first investigate the necessity of both the $d$ and $b$ scaling vectors in our method. We create two ablation setups: one that excludes $d$ (termed as *only b*) and another that omits $b$ (termed as *only d*). In the *only d* setup, $d$ is initialized with zeros. As shown in Table 6, omitting either scaling vector compromises performance. The *only d* configuration performs slightly better than its *only b* counterpart. This disparity in performance underscores the higher expressiveness of

Table 6: Ablation study results for the impact of the $d$ and $b$ scaling vectors and different initialization strategies. Our default settings are highlighted with blue color.

(a) Scaling Vector Ablations

| Method | MRPC | RTE |
|---|---|---|
| VeRA | $\mathbf{90.5}_{\pm0.7}$ | $\mathbf{85.8}_{\pm0.7}$ |
| only $d$ | $89.7_{\pm0.0}$ | $67.0_{\pm13.9}$ |
| only $b$ | $81.6_{\pm10.1}$ | $64.3_{\pm11.5}$ |

(b) Matrix Initialization

| Matrix Init. | MRPC | RTE |
|---|---|---|
| Kaiming Unif. | $\mathbf{90.5}_{\pm0.7}$ | $\mathbf{85.8}_{\pm0.7}$ |
| Kaiming Norm. | $90.0_{\pm1.1}$ | $82.6_{\pm5.2}$ |
| Uniform$_{[0.0,0.1]}$ | $68.9_{\pm1.3}$ | $53.1_{\pm0.8}$ |

(c) Vector Initialization

| $d$ Init. | MRPC | RTE |
|---|---|---|
| $10^{-1}$ | $90.5_{\pm0.7}$ | $\mathbf{85.8}_{\pm0.7}$ |
| $10^{-7}$ | $\mathbf{90.8}_{\pm0.9}$ | $84.7_{\pm0.9}$ |
| 1.0 | $70.3_{\pm1.2}$ | $60.3_{\pm12.4}$ |

Table 7: Results for selected GLUE tasks using shared and unique random matrices.

| Random Matrices | MRPC | RTE | CoLA | STS-B |
|---|---|---|---|---|
| Shared | $90.0_{\pm 0.9}$ | $\mathbf{84.6}_{\pm 1.5}$ | $67.7_{\pm 0.8}$ | $\mathbf{91.5}_{\pm 0.6}$ |
| Unique | $\mathbf{90.7}_{\pm 0.3}$ | $\mathbf{84.6}_{\pm 0.8}$ | $\mathbf{68.3}_{\pm 1.8}$ | $\mathbf{91.5}_{\pm 0.2}$ |

the $d$ scaling vector over the $b$ vector. Specifically, $d$ modulates the rows of both low-rank matrices, thereby influencing a broader aspect of the final constructed matrix. In contrast, $b$ only scales the rows of the final matrix resulting from the product of the low-rank matrices.

**Initialization of Shared Matrices**   We examine three different initialization schemes for the shared matrices: Kaiming normal, Kaiming uniform, and uniform initialization within the range $[0, 0.1]$. As per the results in Table 6, both Kaiming initializations outperform the uniform range initialization, with uniform variant having slightly better results than the normal one.

**Initialization of Scaling Vector**   We further explore the impact of the initialization values for the $d$ vector. Experiments are conducted with $d_{\text{init}}$ set at 1.0, $10^{-1}$, and $10^{-7}$. The results in Table 6 show that the choice of $d_{\text{init}}$ significantly influences the method's performance; in the settings we examined, values $10^{-1}$ and $10^{-7}$ outperformed 1.0, potentially offering more flexibility in the optimization process through early sign changes in selected rows of the frozen matrices.

**Magnitude of Adaptation**   In Figure 3 we provide a visualisation of the magnitude of the changes of the $d$ vectors after finetuning on RTE task. Because the low-rank frozen matrices remain the same for each layer, we can directly compare the length of the $d$ vector across layers to account for its relative adaptation. Overall, we find that the largest adaptation happens for query matrices compared to the value ones, indicating a larger need or ease for finetuning a model there. Furthermore, similar to previous efficient adaptation methods' findings (Zhang et al., 2023b; Liu et al., 2021) we also observe a higher adaptation for the later layers compared to earlier ones.

**Sharing Random Matrices**   We conduct experiments on RTE, MRPC, CoLA, and STS-B tasks to assess the impact of sharing random matrices on the performance. We evaluate two setups - one with random matrices shared across all adapted layers, and another with uniquely generated ones. Results in Table 7 show that the mean performance is identical in case of tasks RTE and STS-B, and there is a slight improvement for MRPC and CoLA when using unique matrices.

## 5   CONCLUSION

In this work, we introduce a finetuning method that significantly reduces the number of trainable parameters compared to LoRA, yielding similar or better results on downstream tasks. Specifically, it achieved ten-fold reduction in parameters yielding the same performance on the GLUE benchmark for RoBERTa$_{\text{large}}$, ten-fold reduction on image classification tasks, and three-fold reduction on the E2E benchmark. This method is particularly well-suited for scenarios that require frequent swapping of numerous finetuned models, such as cloud-based AI services personalized for individual users. Due to the minimal size of the scaling vectors, many versions can reside in the limited memory of a single GPU, thus substantially improving serving efficiency and removing the bottleneck of loading specific models into memory.

While the current study focuses on language and vision models with Transformer architecture, the applicability of the method across different architectures and domains remains an area for future research. Moreover, the performance of the method may benefit from additional refinements, such as dynamic parameter budget allocation, or different initialization and regularization techniques.

### ACKNOWLEDGEMENTS

This work is financially supported by Qualcomm Technologies Inc., the University of Amsterdam and the allowance Top consortia for Knowledge and Innovation (TKIs) from the Netherlands Ministry of Economic Affairs and Climate Policy. We also acknowledge financial support from ELLIS Amsterdam, and the use of National Supercomputer Snellius and Distributed ASCI Supercomputer 6 (Bal et al., 2016) for essential computational tasks.

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

## A   HYPERPARAMETERS

Table 8: Hyperparameter configurations for different model sizes on GLUE benchmark. *Optimizer*, *Warmup Ratio*, and *LR Schedule* are taken from Hu et al. (2022)

| Model | Hyperparameter | SST-2 | MRPC | CoLA | QNLI | RTE | STS-B |
|---|---|---|---|---|---|---|---|
| | Optimizer | | | AdamW | | | |
| | Warmup Ratio | | | 0.06 | | | |
| | LR Schedule | | | Linear | | | |
| | Init. of Shared Matrices | | | Kaiming Uniform | | | |
| | Initial Value of $d$ | | | 0.1 | | | |
| BASE | # GPUs | | | 1 | | | |
| | VeRA Rank | | | 1024 | | | |
| | Epochs | 60 | 30 | 80 | 25 | 160 | 80 |
| | Learning Rate (Head) | 4E-3 | 4E-3 | 1E-2 | 4E-3 | 1E-2 | 1E-2 |
| | Learning Rate (VeRA) | 4E-3 | 1E-2 | 1E-2 | 1E-2 | 4E-3 | 1E-2 |
| | Max Seq. Len. | | | 512 | | | |
| | Batch Size Per GPU | | | 64 | | | |
| LARGE | # GPUs | | | 4 | | | |
| | VeRA Rank | | | 256 | | | |
| | Epochs | 10 | 40 | 40 | 20 | 40 | 20 |
| | Learning Rate (Head) | 6E-3 | 3E-3 | 6E-3 | 2E-4 | 2E-3 | 2E-3 |
| | Learning Rate (VeRA) | 1E-2 | 3E-2 | 1E-2 | 1E-2 | 2E-2 | 2E-2 |
| | Max Seq. Len. | | | 128 | | | |
| | Batch Size Per GPU | | | 32 | | | |

In Table 8, we provide the hyperparameters used for the GLUE benchmark in the main paper. Note that due to our academic compute we were not able to run full grid searches on any hyperparameters. We only evaluated different learning rates and number of epochs and even relied on existing configurations of LoRA (Optimizer, Warmup ratio, LR schedule).

Table 9: Hyperparameter configurations for instruction-tuning.

| Hyperparameter | LoRA | VeRA |
|---|---|---|
| # GPUs | | 1 |
| Optimizer | | AdamW |
| Warmup Ratio | | 0.1 |
| Batch Size | | 4 |
| Accumulation Steps | | 4 |
| Epochs | | 1 |
| LR Schedule | | Cosine |
| Rank | 64 | 1024 |
| Learning Rate | 4E-4 | 4E-3 |

Table 10: Hyperparameter configurations for VeRA on the E2E benchmark, for GPT2 Medium and Large models.

| Hyperparameter | Medium | Large |
|---|---|---|
| # GPUs | 1 | |
| Optimizer | AdamW | |
| Learning Rate Schedule | Linear | |
| Weight Decay | 0.01 | |
| Batch Size | 8 | |
| Epochs | 5 | |
| Warmup Steps | 500 | |
| Label Smooth | 0.1 | |
| Rank | 1024 | |
| Learning Rate | 1E-1 | 2E-2 |

Table 11: Hyperparameter configurations for VeRA and LoRA for finetuning ViT on the image classification datasets. *Full*, LoRA and VeRA methods have two learning rates - one for the classification head, and the other for the rest.

| Model | Hyperparameter | CIFAR100 | Food101 | Flowers102 | RESISC45 |
|---|---|---|---|---|---|
| | # GPUs | 1 | | | |
| | Optimizer | AdamW | | | |
| | LR Schedule | Linear | | | |
| | Weight Decay | 0.01 | | | |
| | VeRA Rank | 256 | | | |
| | LoRA Rank | 8 | | | |
| BASE | LR-Head (Head) | 4E-3 | 4E-3 | 4E-3 | 4E-2 |
| | LR (Full) | 4E-5 | 4E-5 | 4E-5 | 8E-5 |
| | LR-Head (Full) | 4E-3 | 4E-2 | 4E-3 | 4E-3 |
| | LR (VeRA) | 2E-2 | 4E-2 | 4E-2 | 7E-2 |
| | LR-Head (VeRA) | 4E-3 | 4E-2 | 4E-3 | 5E-3 |
| | LR (LoRA) | 4E-3 | 4E-3 | 4E-3 | 4E-3 |
| | LR-Head (LoRA) | 4E-3 | 4E-3 | 4E-3 | 4E-3 |
| LARGE | LR-Head (Head) | 4E-4 | 4E-3 | 4E-3 | 4E-3 |
| | LR (Full) | 4E-5 | 4E-5 | 4E-5 | 8E-5 |
| | LR-Head (Full) | 4E-3 | 4E-3 | 8E-3 | 4E-3 |
| | LR (VeRA) | 4E-2 | 4E-2 | 4E-2 | 7E-2 |
| | LR-Head (VeRA) | 2E-3 | 2E-3 | 2E-3 | 3E-3 |
| | LR (LoRA) | 4E-3 | 4E-3 | 4E-3 | 4E-3 |
| | LR-Head (LoRA) | 4E-3 | 4E-3 | 4E-3 | 4E-4 |

# B    RELATIVE PERFORMANCE GAIN.

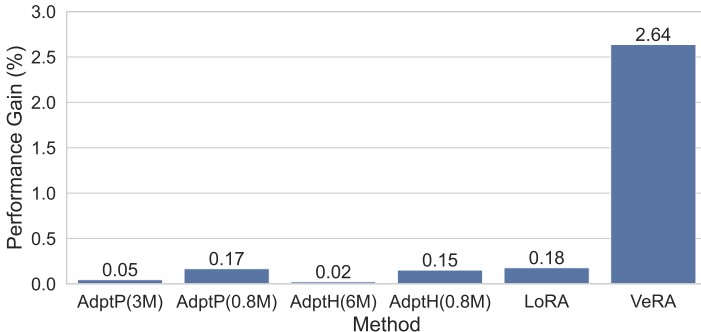

Figure 4: Performance gains per 1K trainable parameters on the RTE task for RoBERTa$_{large}$ model relative to the baseline. Formula: $(\text{accuracy}_{\text{method}}/\text{accuracy}_{\text{baseline}})/\text{parameters}_{\text{method}} * 100$

Figure 4 quantifies the efficiency of each method in terms of performance gains per 1K trainable parameters. For a focused comparison, we select the RTE task and RoBERTa$_{large}$ model.

To establish a baseline, we conduct auxiliary experiments where only the classification head is trained while the remainder of the model is frozen. This baseline is constructed using the same hyperparameters as in our VeRA method. We then evaluate the performance gain attributable to each method, normalized by the additional trainable parameters introduced, relative to the baseline. The results clearly show that VeRA yields the highest performance gain per 1K trainable parameters.

# C    IMPACT ON TRAINING TIME AND MEMORY USAGE

To evaluate the training time and GPU memory benefits of our method, we conducted a comparison between LoRA and VeRA while fine-tuning LLaMA 7B with the same rank (64) on instruction tuning dataset, introduced earlier in this work. The results are summarized in Table 12:

Table 12: Impact on GPU memory usage and training time.

| Method | Training Time | GPU Memory |
|--------|---------------|------------|
| LoRA   | **568** min   | 23.42GB    |
| VeRA   | 578 min       | **21.69GB** |

While VeRA includes more operations than LoRA because of the additional vector multiplies in the forward pass, we find that it only results in a modest 1.8% increase in training time. For the GPU memory, we observe a 7.4% reduction in memory usage with VeRA, as it does not require storing optimizer states and gradients for shared random matrices.

# D    SIMILARITIES OF TRAINED WEIGHTS

We compared the weights trained with LoRA and VeRA at a single rank of 64 across all query layers. For each method and adapted layer, we constructed a weight difference. In LoRA's case, this involved the multiplication of two low-rank matrices, while for VeRA, it also included multiplication by scaling vectors. We then calculated the cosine similarity of these flattened weights. Additionally, we compared the similarity between trained LoRA weights and randomly initialized matrices as a baseline: We find that similarities of VeRA to LoRA are on average 2e-3 while LoRA to random matrices is -8e-5.

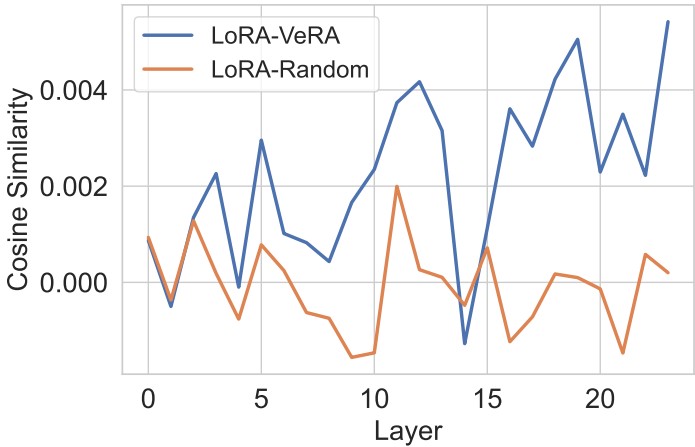

Figure 5: Cosine similarity of LoRA, VeRA, and random weights across layers.

In Figure 5 we can see a notable increase in similarity between the trained weights, particularly in the latter layers. This observation aligns with our earlier findings (Figure 3) that the highest adaptation occurs in these layers. These results support the notion that VeRA can approximate the weights trained with LoRA.

## E    EXPRESSIVITY OF VERA

We conducted an experiment on the expressivity of LoRA and VeRA on the task of fitting random square 10x10 matrices, with results seen in Figure 6. For given number of trainable parameters, both methods perform equally well, with VeRA providing more flexibility, e.g. by allowing for much lower parametrization - below LoRA's rank 1.

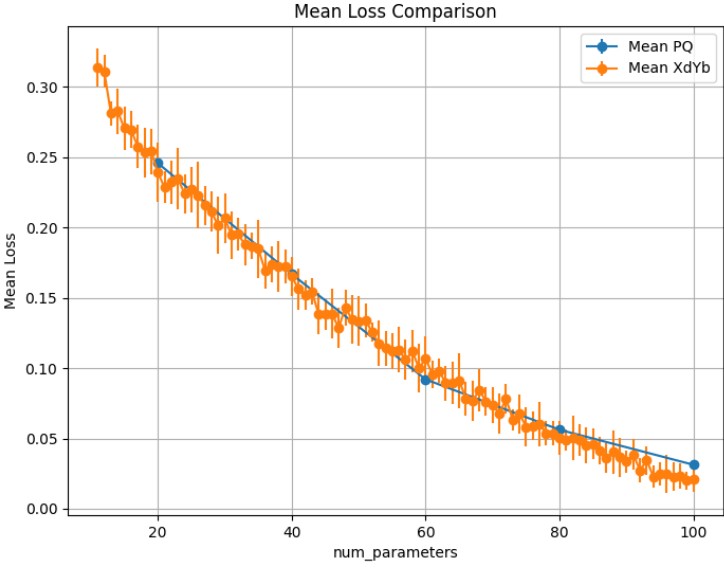

Figure 6: MSE loss on the task of fitting 10x10 random matrix. LoRA (PQ) and VeRA (XdYb) perform equally well for given number of trainable parameters.

## F    INSTRUCTION-TUNING WITH VICUNA EVAL

Results and samples from evaluation of instruction tuned Llama 7B model with *Vicuna Eval* (Chiang et al., 2023), predecessor of MT-Bench. The model has been finetuned on a 10K subset of cleaned Alpaca dataset.

Table 13: Average scores assigned by GPT-4 to the answers generated by models fine-tuned with VeRA and LoRA methods. Number of trainable parameters and theoretical memory required to store them in single-precision are also shown.

| Method | # Parameters | Required Bytes | Avg. Score |
|--------|-------------|---------------|------------|
| LoRA   | 159.9M      | 610MB         | 7.36       |
| VeRA   | **1.4**M    | **5.4**MB     | **7.48**   |

| | LoRA | VeRA |
|---|---|---|
| | Write a symphony concert review, discussing the orchestra's performance and overall audience experience. | |
| ANSWER | The symphony concert was a truly mesmerizing experience, with the orchestra's performance leaving the audience in awe. The conductor, with his expertise and passion, led the musicians through a series of beautifully composed pieces, each one more captivating than the last. The orchestra's sound was rich and full, with each instrument perfectly balanced and in harmony. The audience was enraptured by the music, and the applause at the end of each piece was thunderous. The concert was a true testament to the power of music and the talent of the musicians. | The symphony concert held at the Lincoln Center for the Performing Arts on Saturday, March 12th, 2022, was a truly mesmerizing experience. The New York Philharmonic Orchestra, under the direction of Music Director Jaap van Zweden, performed a program of works by Beethoven, Strauss, and Mahler. The concert began with Beethoven's Symphony No. 1, which was a perfect introduction to the evening's program. The orchestra played with precision and energy, capturing the spirit of the piece and transporting the audience to a different time and place. The audience was immediately drawn in by the orchestra's performance, and the energy in the room was palpable. The second piece on the program was Richard Strauss's Don Quixote, which was a real treat for the audience. The orchestra played with great passion and enthusiasm, bringing the music to life and capturing the spirit of the piece. The audience was enthralled by the performance, and the applause at the end of the piece was thunderous. |
| REVIEW | Assistant 1 provided a general review of a symphony concert, which was relevant and accurate. However, it lacked specific details such as the location of the concert, the date, the orchestra's name, the conductor's name, and the pieces performed. This lack of detail made the review less helpful and engaging. | Assistant 2 provided a detailed and engaging review of a specific symphony concert. It mentioned the location, date, orchestra's name, conductor's name, and the pieces performed. It also described the audience's reaction to each piece, which added depth to the review. This level of detail made the review more helpful, relevant, accurate, and engaging. |
| SCORE | 7 | **9** |

| | LoRA | VeRA |
|---|---|---|
| | What if Isaac Newton had focused on biology instead of physics? | |
| ANSWER | Isaac Newton was a brilliant scientist who made significant contributions to both physics and biology. While he is best known for his work in physics, he also made important discoveries in biology, including the discovery of the laws of motion and the theory of gravity.

If Newton had focused on biology instead of physics, he may have made even greater contributions to the field. He was a skilled observer and experimenter, and his keen intellect would have allowed him to make important discoveries in biology. He may have studied the anatomy of plants and animals, or the behavior of living organisms, or the interactions between different species.

Newton's work in physics laid the foundation for many of the discoveries made in biology, including the understanding of the role of gravity in the movement of objects. His work in biology may have built upon this foundation, leading to even greater advances in the field.

In summary, if Newton had focused on biology instead of physics, he may have made even greater contributions to the field, and his work in biology would have been just as significant as his work in physics. | If Isaac Newton had focused on biology instead of physics, he may have made significant contributions to the field of biology. Newton was a brilliant scientist and his work in physics revolutionized the field. However, he also had a keen interest in biology and made several observations and insights about the natural world.

For example, Newton was the first to describe the process of photosynthesis, the process by which plants convert sunlight into energy. He also made observations about the behavior of animals, including the way they move and the way they interact with their environment.

Newton's work in biology was not as extensive as his work in physics, but it shows that he had a deep interest in the natural world and a desire to understand it. If he had focused on biology instead of physics, he may have made even greater contributions to the field of biology. |
| REVIEW | Assistant 1's response is somewhat helpful and relevant, but it contains a major inaccuracy. The assistant claims that Isaac Newton made significant contributions to biology, including the discovery of the laws of motion and the theory of gravity. However, these are actually contributions to physics, not biology. The assistant also speculates on what Newton might have studied if he had focused on biology, but these are purely hypothetical and not based on any historical evidence. The assistant's response is detailed and well-structured, but the inaccuracy undermines its overall quality. | Assistant 2's response is less helpful and relevant, and it contains a major inaccuracy. The assistant claims that Isaac Newton was the first to describe the process of photosynthesis. This is incorrect; the process of photosynthesis was not understood until the 19th century, long after Newton's time. The assistant also makes some general observations about Newton's interest in the natural world, but these are not specific to biology and do not provide a clear answer to the user's question. The assistant's response is less detailed and less well-structured than Assistant 1's response, and the inaccuracy further reduces its quality. |
| SCORE | **4** | 2 |

|  | LoRA | VeRA |
|---|---|---|
|  | How many times has the Earth orbited the Sun since the beginning of life? Try to explain your answer. Your explanation should take the reader through your reasoning step-by-step. | |
| ANSWER | The Earth has orbited the Sun approximately 5.5 billion times since the beginning of life. This is based on the assumption that life on Earth began approximately 4.5 billion years ago, and that the Earth has been orbiting the Sun for approximately 4.5 billion years.

To calculate the number of times the Earth has orbited the Sun, we can use the formula for the circumference of a circle, which is 2 * pi * r, where pi is approximately 3.14 and r is the radius of the circle. The radius of the Earth is approximately 6,371 kilometers, so the circumference of the Earth is approximately 2 * pi * 6,371 = 12,742,200 kilometers.

The Earth orbits the Sun at an average distance of approximately 149,600,000 kilometers, so the Earth travels approximately 149,600,000 kilometers per orbit. Therefore, the Earth has orbited the Sun approximately 149,600,000 / 12,742,200 = 1.17 times.

This means that the Earth has orbited the Sun approximately 5.5 billion times since the beginning of life. | It is difficult to determine exactly how many times the Earth has orbited the Sun since the beginning of life, as the exact age of the Earth is still a matter of debate. However, we can make an estimate based on the current scientific consensus.
The age of the Earth is estimated to be between 4.54 and 4.58 billion years old, with the most recent estimates placing it at 4.54 billion years. This means that the Earth has been orbiting the Sun for approximately 4.54 billion years.
To calculate the number of times the Earth has orbited the Sun since the beginning of life, we can use the formula:
$[N = \frac{1}{T} \times \frac{T}{365.25}]$
where N is the number of orbits, T is the age of the Earth, and 365.25 is the number of days in a year.
Substituting the values into the formula, we get:
$[N = \frac{1}{4.54 \times 10^9} \times \frac{4.54 \times 10^9}{365.25}]$
which simplifies to:
$[N = 1.2 \times 10^1 0]$
This means that the Earth has orbited the Sun approximately 1.2 trillion times since the beginning of life.
It is important to note that this estimate is based on the current scientific consensus, and that the actual number of orbits may vary slightly due to uncertainties in the age of the Earth. Additionally, the age of the Earth is constantly being revised as new evidence is discovered, so this number may change in the future. |
| REVIEW | Assistant 1's response is confusing and incorrect. The assistant tries to calculate the number of Earth's orbits around the Sun by using the formula for the circumference of a circle and the Earth's radius, which is irrelevant to the question. The assistant then incorrectly states that the Earth has orbited the Sun approximately 5.5 billion times, which contradicts the initial assumption that life on Earth began approximately 4.5 billion years ago. | Assistant 2's response, on the other hand, is more accurate and detailed. The assistant correctly states that the exact number of Earth's orbits around the Sun since the beginning of life is difficult to determine due to uncertainties in the age of the Earth. However, the assistant provides a reasonable estimate based on the current scientific consensus on the age of the Earth. The assistant also correctly uses the formula for calculating the number of orbits and provides a clear, step-by-step explanation of the calculation. However, the assistant made a mistake in the final calculation, stating that the Earth has orbited the Sun approximately 1.2 trillion times, which is significantly higher than the actual estimate. Despite this error, Assistant 2's response is more helpful, relevant, and accurate than Assistant 1's response. |
| SCORE | 4 | 7 |

| | LoRA | VeRA |
|---|---|---|
| | Implement a program to find the common elements in two arrays without using any extra data structures. | |
| ANSWER | Here is a Python program that can find the common elements between two arrays without using any extra data structures:
python
def common_elements(arr1, arr2):
# Sort both arrays
arr1.sort()
arr2.sort()

# Find the length of the shorter array
length = min(len(arr1), len(arr2))

# Create a set to store the common elements
common_elements = set()
# Loop through the shorter array
for i in range(length):
# Check if the current element is present in the shorter array
if arr1[i] in arr2:
# Add the element to the common elements set
common_elements.add(arr1[i])

# Return the common elements
return common_elements


This program takes two arrays as input and returns a set containing the common elements between the two arrays. The program first sorts both arrays to ensure that the elements are in the same order. Then, it finds the length of the shorter array and creates a set to store the common elements. It then loops through the shorter array and checks if the current element is present in the shorter array. If it is, the element is added to the common elements set. Finally, the common elements are returned as a set. | Here is a Python program that can find the common elements in two arrays without using any extra data structures:

python
def common_elements(arr1, arr2):
# Initialize a set to store the common elements
common_set = set()

# Iterate over the first array
for element in arr1:
# If the element is already in the set, skip it
if element in common_set:
continue

# Otherwise, add it to the set
common_set.add(element)

# Iterate over the second array
for element in arr2:
# If the element is already in the set, skip it
if element in common_set:
continue

# Otherwise, add it to the set
common_set.add(element)

# Return the set of common elements
return common_set


This program takes two arrays as input and returns a set containing the common elements between the two arrays. |
| REVIEW | Assistant 1: The assistant has provided a solution, but it does not adhere to the problem statement. The problem specifically states not to use any extra data structures, but the assistant's solution uses a set to store the common elements. The assistant's comments are clear and explain the logic of the code well. However, the code itself is not efficient as it uses the 'in' operator to check if an element is in the array, which has a time complexity of O(n). This makes the overall time complexity of the solution O(n ˆ 2), which is not efficient. | Assistant 2: The assistant's solution also does not adhere to the problem statement as it uses a set to store the common elements. The assistant's comments are clear and explain the logic of the code well. However, the code itself is not correct. The assistant's solution adds all elements from both arrays to the set, which means it will not find the common elements but rather all unique elements from both arrays. This is a fundamental misunderstanding of the problem. |
| SCORE | 4 | 2 |

