# OpenReview forum: "VeRA: Vector-based Random Matrix Adaptation"
_ICLR.cc/2024/Conference — ICLR 2024 poster_

### Official Review · Reviewer_rHtj · 2023-10-26

**Soundness:** 3 good
**Presentation:** 4 excellent
**Contribution:** 4 excellent
**Rating:** 8
**Confidence:** 4

**Summary:**

> **TL;DR:** The proposed ELoRA algorithm achieves a 10x reduction in trainable parameters compared to LoRA, while maintaining performance levels. This paper should be accepted in its current form. Addressing my concerns and questions would improve my score.

The paper introduces an efficient finetuning method, ELoRA, for pretrained language models, addressing the storage challenge when deploying numerous adapted models. ELoRA achieves a 10x reduction in trainable parameters compared to LoRA, while maintaining performance levels, as demonstrated on the GLUE benchmark. It leverages shared random low-rank matrices and small layer-wise trainable scaling vectors to reduce memory usage. This approach is ideal for scenarios requiring multiple finetuned models, such as cloud-based personalized AI services, by improving serving efficiency. The method's potential application in various domains and further optimizations like dynamic parameter allocation are areas for future research.

**Strengths:**

* **S.1.** The paper is well written and is easy to follow. The illustrations and results are informative and clear.
* **S.2.** The proposed ELoRA algorithm seems novel and tackles an important problem.
* **S.3.** The experiments are conducted on multiple datasets and architectures with 5 different random seeds. ELORA achieves high accuracy while requiring significantly less parameters compared to existing algorithms.
* **S.4.** The experiments are thorough and an ablation study is conducted.

**Weaknesses:**

* **W.1.** The experiments are conducted solely on NLP tasks with LLMs. Adding experiments on different domains such as diffusions on CV would help.
* **W.2.** The paper does not provide a theoretical analysis on the ELoRA algorithm or its limitations.

**Questions:**

* **Q.1.** In Table 2 for the RTE dataset the LoRA algorithm achieves significantly higher results on the RoBERTa-base model compared to the the other algorithms. Is this a mistake?
* **Q.2.** Why are the PEFT algorithms achieving higher accuracy compared FT on the RTE dataset? Is this just overfitting?

---

> ### Author Response · Authors · 2023-11-18
> **Response to rHtj**
>
> Thank you for your constructive feedback and detailed review. We have taken your suggestions and expanded our experiments to encompass not just NLP tasks but also applications in computer vision, demonstrating the versatility of our approach. Our responses below address each of your points, with parts on the vision experiments and the theoretical analysis moved to the common reply.
>
>
> >Adding experiments on different domains such as diffusions on CV would help.
>
> Please see our response in the "Experiments on vision models" part of the [common reply](https://openreview.net/forum?id=NjNfLdxr3A&noteId=xpnpKQguRK).
>
> >Theoretical analysis on the ELoRA algorithm or its limitations
>
> Please see our response in the "Theoretical framework/expressivity of ELoRA" part of the [common reply](https://openreview.net/forum?id=NjNfLdxr3A&noteId=xpnpKQguRK).
>
>
>
>
> >In Table 2, on the RTE task LoRA is significantly better than other algorithms. Is this a mistake?
>
> Results reported for LoRA are taken from the paper that introduced the method [1]. Our reproduced results are _close_ to this value, which may indicate that it's not a mistake.
>
>
> >Why are the PEFT algorithms achieving higher accuracy compared FT on the RTE dataset? Is this just overfitting?
>
> Indeed, overfitting may be the reason of lower performance of FT compared to PEFT methods. It may also explain higher performance of ELoRA on certain tasks, as it comes with highly reduced number of trainable parameters, potentially acting as regularisation.
>
>
> **References**
> - [1] LoRA: Low-Rank Adaptation of Large Language Models, Edward J. Hu et al., 2021

---

### Official Review · Reviewer_iJDt · 2023-11-01

**Soundness:** 3 good
**Presentation:** 2 fair
**Contribution:** 2 fair
**Rating:** 5
**Confidence:** 2

**Summary:**

This paper propose a way of reparametrization of the low-rank matrices in a LoRA method, by freezing a single pair of randomly initialized matrices shared across all adapted layers, and introducing trainable scaling vectors that allow for layer-wise adaptation.

**Strengths:**

- Introduced a simple but elegant way to reduce trainable parameters in a LoRA. This effectively reduces storage cost due to introduced parameters for different downstream tasks.

**Weaknesses:**

- Performance can be hurted using this method, even the evaluation is only on a limited set of GLUE benchmarks.

- The method should be evaluated on generative tasks as well, including translation and question answering. Classification task and summarization task are known to be more resilient to the lose of model capacity or trainable parameters.

- The paper is not well written. Figure 3 and Figure 4 are each on a single but different task. Why not reporting a mean score on GLUE?

- The paper lacks explanation on why Table 3, with fewer tunable parameters, ELoRA has better instruction finetuning scores than LoRA. It is a bit of counterintuitive.

**Questions:**

- Are there any theoretical proof that random matrices with scaling vectors would work better than LoRA?

- Consider evaluating the method on Generative tasks?

---

> ### Author Response · Authors · 2023-11-18
> **Response to iJDt**
>
> Thank you for your feedback and time to review our paper. Based on this, we have further conducted more experiments on generative E2E task and MT-Bench. Please find our detailed responses to each of your points raised below:
>
> >Performance can be hurted using this method, even the evaluation is only on a limited set of GLUE benchmarks.
>
> The primary objective of our method is to significantly reduce the number of trainable parameters, and therefore storage requirements, with *comparable* performance to other methods. As detailed in Figure 3, our analysis acknowledges the parameter-performance trade-off inherent in both LoRA and ELoRA methods. While in theory, ELoRA _could_ perform worse than LoRA, in practice we see that with a sufficient number of trainable parameters, ELoRA's performance aligns with that of LoRA, despite our method using much less parameters.
>
> In addition, we have also increased the types of evaluations by comparing our method on E2E generation  (see answer below) and on instruction following tasks - additionally to the results reported in the initial version of our paper, we are introducing evaluation with MT-Bench, a well-established instruction-following benchmark. Similarly to the initial experiment, we evaluate model finetuned on the *alpaca-cleaned* dataset, this time taking the full training set (40k+ samples), instead of 10k subset. Results are shown in Table 1.
>
>
> **Table 1 Instruction tuning of more models and evaluated on MT-Bench**
>
> | Model        | Method | Parameters | Score |
> |--------------|--------:|------------:|-------|
> | Llama 13B    |       | -          | 2.61  |
> ||||
> | Llama 7B     | LoRA   | 159.9M     | 5.03  |
> | Llama 7B     | ELoRA  | 1.6M       | 4.77  |
> ||||
> | Llama 13B    | LoRA   | 250.3M     | 5.31  |
> | Llama 13B    | ELoRA  | 2.4M       | 5.22  |
> ||||
> | Llama2 7B    | LoRA   | 159.9M     | 5.19  |
> | Llama2 7B    | ELoRA  | 1.6M       | 5.08  |
> ||||
> | Llama2 13B   | LoRA   | 250.3M     | 5.77  |
> | Llama2 13B   | ELoRA  | 2.4M       | 5.93  |
>
> From this table, we make the following observations:
>  - Base models do not perform well compared to instruction tuned versions (scores of 2.6 vs 5.3)
>  - ELoRA generally closely matches LoRA's performance. In particular for the larger model sizes, such as Llama 13B the scores are very similar (5.22 vs 5.31) and for Llama2 13B ELoRA even outperforms LoRA (5.93 vs 5.77).
>
> >Consider evaluating the method on Generative tasks?
>
> We have conducted additional experiments on a generative E2E task with GPT2, medium and large variants, finetuned with LoRA and ELoRA methods, adhering to the setup described in the original LoRA paper [1]. For LoRA we use hyperparameters reported in [1], and for ELoRA we select a rank equal to 1024 and tune the learning rate. Results after 5 epochs are reported in the following table:
>
> | Method               | Parameters | BLEU  | NIST   | METEOR | ROUGE_L | CIDEr  |
> |---------------------|------------|-------|--------|--------|---------|--------|
> | LoRA (GPT2-M)        | 0.35M      | 68.92 | 8.69 | 46.42  | 71.34   | **2.51** |
> | ELoRA (GPT2-M)    | **0.10M**     | **70.08** | **8.81** | **46.55**  | **71.45**   | 2.50 |
> |||||
> | LoRA (GPT2-L)         | 0.77M      | 70.14 | 8.84 | 46.68  | **71.85**   | 2.52 |
> | ELoRA (GPT2-L)     | **0.17M**     | **70.28** | **8.85** | **46.89**  | 71.63   | **2.54** |
>
> As we can see in the table, our method **performs on-par or better on E2E-NLG, despite using only 22%-28% of the number of trainable parameters of LoRA**. We thank the reviewer for suggesting this additional experiment and will add these results to the paper.
>
>
>
> >Figure 3 and Figure 4 are each on a single but different task.
>
> While Figures 3 and 4 are independent and address different aspects, we agree that consistency could enhance the clarity of our presentation.   Consequently, we will update Figure 4 to showcase data from the RTE task, making it consistent with Figure 3.
>
> >Why not report a mean score on GLUE?
>
> In terms of reporting on the GLUE score, our methodology is in line with that of the LoRA paper [1], which opts to report median values. This approach enables a straightforward comparison between our results and those reported in the LoRA paper and is more robust to outliers.
>
>
> >Why is ELoRA better than LoRA in Table 3?
>
> One possible explanation is that ELoRA is less prone to overfitting and may achieve better results on particular tasks, especially in low-data regime. This is similar to the existing findings of PEFT methods often achieving better results than full finetuning on some tasks.
>
> >Theory about ELoRA
>
> Please see our response in the "Theoretical framework/expressivity of ELoRA" part of the [common reply](https://openreview.net/forum?id=NjNfLdxr3A&noteId=xpnpKQguRK).
>
> **References**
> - [1] LoRA: Low-Rank Adaptation of Large Language Models, Edward J. Hu et al., 2021

---

> > ### Author Response · Authors · 2023-11-22
> >
> > We sincerely hope that our answers have met your expectations and satisfactorily addressed your inquiries. Please let us know if there are any further questions until the discussion period ends today! Finally, we would appreciate it if you update your review based on the additional experiments and explanations provided.
> >
> > Thank you for your time,
> > Authors

---

> > > ### Comment · Reviewer_iJDt · 2023-12-01
> > > **Thank you for the responses.**
> > >
> > > The authors' responses mostly addressed my questions. Please add the generative task scores in the final paper. I'd like to increase my score accordingly.

---

### Official Review · Reviewer_JAXk · 2023-11-06

**Soundness:** 4 excellent
**Presentation:** 4 excellent
**Contribution:** 4 excellent
**Rating:** 8
**Confidence:** 3

**Summary:**

This paper proposes a novel parameter-efficient LLM finetuning algorithm which is considerably more parameter-efficient than LoRA. The idea is to use 2 random matrices shared across all of the layers in the LLM, but to use two sets of diagonal scaling matrices to modulate the shared random matrices differently for each layer. The results indicate that the proposed method requires roughly an order of magnitude less parameters while maintaining or exceeding the performance of LoRA.

**Strengths:**

1) The paper is very easy to read
2) The method is simple and easy to understand
3) The experimental results are extensive. All claims are backed up by experimental evidence. Ablation experiments give further insights

**Weaknesses:**

I don't really see any weaknesses in this paper

**Questions:**

1) Have you tried extending this method to finetuning large image models? What about large multimodal models?
2) What are the training time benefits of your method? Is there a GPU memory benefit? Is there a latency benefit? Does your method train faster than LoRA because it has less parameters?

---

> ### Author Response · Authors · 2023-11-18
> **Response to JAXk**
>
> Thank you for your positive feedback and your suggestions to further improve the paper! Most importantly, we have now conducted experiments using Vision Transformers and shown benefits of ELoRA in terms of GPU memory. Please see below for our responses to your questions.
>
> >Have you tried extending this method to finetuning large image models?
>
> Please see our response in the "Experiments on vision models" part of the [common reply](https://openreview.net/forum?id=NjNfLdxr3A&noteId=xpnpKQguRK).
>
> >What are the training time benefits of your method? Is there a GPU memory benefit? Is there a latency benefit? Does your method train faster than LoRA because it has less parameters?
>
> First, regarding inference latency, both LoRA and ELoRA exhibit no differences. In both methods, the trainable matrices and vectors can be merged into the original weights, ensuring that the inference latency remains on par with the base model. There would, however, be benefits for our lighter-weight adapters, e.g. when serving many of them concurrently [1].
>
> To evaluate the training time and GPU memory benefits of our method, we conducted a comparison between LoRA and ELoRA while fine-tuning LLaMA 7B with the same rank (64). The results are summarized in the table below:
>
> | Resource         | LoRA    | ELoRA (ours)  |
> |------------------|---------|---------|
> | Training Time    | 568 min | 578 min |
> | GPU Memory Usage | 23.42GB | 21.69GB |
>
> While ELoRA includes more operations than LoRA because of the additional vector multiplies in the forward pass, we find that it only results in a modest 1.8% increase in training time.
> For the GPU memory, we observe a **7.4% reduction in VRAM usage with ELoRA**, as it does not require storing optimizer states and gradients for shared random matrices.
> We will add this to the paper and again thank the reviewer for raising this point.
>
>
> **References**
> - [1] Sheng et al. S-LoRA: Serving Thousands of Concurrent LoRA Adapters. Nov 2023, ArXiv

---

### Official Review · Reviewer_Bg6K · 2023-11-09

**Soundness:** 3 good
**Presentation:** 3 good
**Contribution:** 3 good
**Rating:** 8
**Confidence:** 3

**Summary:**

This work proposes ELoRA, a new method for parameter-efficient finetuning that trains only the scaling vectors for low-rank parametrized versions of linear layers. The proposed method reaches the same quality as LoRA on a variety of benchmarks while having more than 10 times fewer trainable parameters.

---
Post-rebuttal update: thank you for the revised version! It addresses most of my concerns, and I am happy to increase the score to 8.

**Strengths:**

* The work proposes a simple idea that works quite well in practice, according to the experiments.
* Authors have a comprehensive evaluation setup, testing ELoRA across several model sizes and datasets and comparing it with baselines.
* The influence of each component of ELoRA is investigated within a detailed ablation study.
* Overall, the paper is well-written and all the contributions are easy to understand.

**Weaknesses:**

* The idea of using random matrices for parameter-efficient finetuning has been proposed previously in the context of text-to-image generation [1]. Although the ideas of the submission and that paper are quite different (also, the paper is quite recent), in my opinion, it would be great to mention LiDB (Lightweight DreamBooth) in the related work section to give the reader additional context.
* My primary concern with respect to the experiments is that the instruction tuning evaluation is performed mostly through GPT-4 queries on a very small dataset. In my opinion, this approach suffers both from the lack of reproducibility (as we know, the model behind the GPT-4 API might change in the future) and a narrow coverage of possible contexts that might result in high variance of the metric. I think that using more established benchmarks (such as MMLU or BIG-Bench) would give a better picture of how instruction-finetuned models perform after PEFT.
* Similarly, I think that broader evaluation of instruction tuning should include a larger set of models than just LLaMA-2. Personally, I would move Table 4 and Table 5 to the appendix (they take a lot of space and are not directly related to the topic of the paper) and replace that with additional experiments, ideally for models with higher parameter counts.

[1] HyperDreamBooth: HyperNetworks for Fast Personalization of Text-to-Image Models. Nataniel Ruiz, Yuanzhen Li, Varun Jampani, Wei Wei, Tingbo Hou, Yael Pritch, Neal Wadhwa, Michael Rubinstein, Kfir Aberman. 2023

**Questions:**

* Technically, it should be possible to have different random matrices in different layers (the cost of storing one random seed per layer should be negligible), which might increase the capacity of the model even further. I wonder if you have explored this idea in preliminary experiments? It is quite surprising to me that a single random structure is sufficient for PEFT across all layers in a Transformer model — perhaps there is something we can infer from that.
* I think that the work would benefit from a bit more analysis behind the reasons of why random projections work that well in the context of parameter-efficient finetuning. At the very least, it might be interesting to compare the structure of the learned weights for LoRA and ELoRA trained with a single rank: intuitively, the latter should approximate the former, but is that true in practice?

---

> ### Author Response · Authors · 2023-11-18
> **Response to Bg6K (part 1)**
>
> Thank you for your feedback and your pointers and questions. Please find our answers and our newly conducted experimental results below.
>
> >It would be great to mention LiDB (Lightweight DreamBooth) in the related work section.
>
>
> Thank you for pointing this out. We acknowledge the paper's relevance and will include it in our updated related work section.
>
>
>
> >Instruction tuning evaluated on a very small dataset.
>
> While using GPT-4 as-a-judge is well-established (e.g. [1,2]), we have now run **additional experiments**: We have evaluated our instruction-tuned models under the well-known multi-turn evaluation, MT-Bench [1], which offers a more robust evaluation compared to the previous method and is one of the key metrics for comparing instruction-tuned methods.
> Similarly to the initial experiment, we evaluate model finetuned on the *alpaca-cleaned* dataset, this time taking the full training set (40k+ samples), instead of 10k subset. As shown in Table 1 below, we see that our model closely matches the performance of LoRA-based finetuning here too, despite using much less parameters.
>
> >Instruction tuning: Models beyond LLaMA-2 7b.
>
> We have further increased the **model diversity**: Our additional experiments include two variants of LLaMA-1 and LLaMA-2, specifically the 7B and 13B models. This demonstrates ELoRA's applicability across different model sizes.
>
> In addition, we now **compare against the baseline model**: We've included results for the base LLaMA 13B model without instruction tuning as a baseline [3].
>
>
>
>
> **Table 1 Instruction tuning of more models and evaluated on MT-Bench**
>
> | Model        | Method | Parameters | Score |
> |--------------|--------:|------------:|-------|
> | Llama 13B    |       | -          | 2.61  |
> ||||
> | Llama 7B     | LoRA   | 159.9M     | 5.03  |
> | Llama 7B     | ELoRA | 1.6M       | 4.77  |
> ||||
> | Llama 13B    | LoRA   | 250.3M     | 5.31  |
> | Llama 13B    | ELoRA  | 2.4M       | 5.22  |
> ||||
> | Llama2 7B    | LoRA   | 159.9M     | 5.19  |
> | Llama2 7B    | ELoRA  | 1.6M       | 5.08  |
> ||||
> | Llama2 13B   | LoRA   | 250.3M     | 5.77  |
> | Llama2 13B   | ELoRA  | 2.4M       | 5.93  |
>
> From this table, we make the following observations:
>  - Base models do not perform well compared to instruction tuned versions (scores of 2.6 vs 5.3)
>  - ELoRA generally closely matches LoRA's performance. In particular for the larger model sizes, such as Llama 13B the scores are very similar (5.22 vs 5.31) and for Llama-v2 13B ELoRA even outperforms LoRA (5.93 vs 5.77).
>
> Thank you for suggesting this. We will add these results to the paper.

---

> > ### Author Response · Authors · 2023-11-18
> > **Response to Bg6K (part 2)**
> >
> > >Effect of different random matrices in different layers
> >
> > We have conducted additional experiments to assess the impact of using unique versus shared random matrices in different layers of a Transformer model. As shown in Table 2, for a rank of 256, our results indicate a slight improvement in performance when using unique random pairs. A reason for this might be that this alleviates the effect of non-optimally initialised shared matrices. Yet, given how small the difference is, this suggests that even this additional slack is not necessary when it comes to finetuning, echoing our findings of comparing LoRA to our method where less parameters did not translate into worse performance.
> >
> > Besides this, opting for unique matrices can lead to increased training times (when re-generating the matrices from seed) or, if these matrices are stored in memory, there will be a minor increase in memory usage. We will add this analysis to the paper.
> >
> >
> >
> > **Table 2 Comparison of shared vs unique matrices**
> > Our experiments were carried out on four GLUE tasks (the fastest ones to compute): STS-B, CoLA, RTE, and MRPC. We employed two setups to compare shared and unique matrices, repeating each experiment five times to calculate the mean results and standard deviation.
> > | Task | Shared Matrices (Mean ± SD) | Unique Matrices (Mean ± SD) |
> > |------|-----------------------------|-----------------------------|
> > | STSB | 91.5 ± 0.6                  | 91.5 ± 0.2                  |
> > | CoLA | 67.7 ± 0.8                  | 68.3 ± 1.8                  |
> > | RTE  | 84.6 ± 1.5                  | 84.6 ± 0.8                  |
> > | MRPC | 90.0 ± 0.9                  | 90.7 ± 0.3                  |
> >
> >
> >
> > >Comparison of the learned weights for LoRA and ELoRA
> >
> >
> > To address your query, we compared the weights trained with LoRA and ELoRA at a single rank of 64 across all *query* layers. For each method and adapted layer, we constructed a _weight difference_. In LoRA's case, this involved the multiplication of two low-rank matrices, while for ELoRA, it also included multiplication by scaling vectors. We then calculated the cosine similarity of these flattened weights. Additionally, we compared the similarity between trained LoRA weights and randomly initialized matrices as a baseline: We find that similarities of ELoRA to LoRA are on average 2e-3  while LoRA to random matrices is -8e-5 (low numbers are due to the high dimensionality).
> >
> >
> > Our findings reveal a notable increase in similarity between the trained weights, particularly in the latter layers. This observation aligns with our earlier findings (referenced as Figure 4 in our paper) that the highest adaptation occurs in these layers. These results support the notion that ELoRA can approximate the weights trained with LoRA. Please also see our response about the theory and expressive power of ELoRA in the [common reply](https://openreview.net/forum?id=NjNfLdxr3A&noteId=xpnpKQguRK).
> >
> >
> > **References**
> > - [1] Judging LLM-as-a-Judge with MT-Bench and Chatbot Arena, Lianmin Zheng et al., 2023
> > - [2] QLoRA: Efficient Finetuning of Quantized LLMs, Dettmers et al., 2023
> > - [3] [MT-Bench Leaderboard](https://huggingface.co/spaces/lmsys/chatbot-arena-leaderboard)

---

### Author Response · Authors · 2023-11-18
**Joint Response**

Here we respond to questions raised by multiple reviewers in more detail.

### Theoretical framework / expressivity of ELoRA (Bg6K, iJDt, rHtj)

Similar to the original LoRA paper, our approach in this paper is predominantly empirical, focusing on demonstrating the practical effectiveness and efficiency of ELoRA. While a comprehensive theoretical analysis of ELoRA is not provided, we do draw upon existing theory and evidence supporting the use of random matrices in similar contexts: random matrices being almost always full-rank (rank r<N matrices form a Lebesque measure zero set in the set of NxN matrices) and the Johnson–Lindenstrauss lemma which shows that random matrix projections approximately keep low-dimensional structures [1] and that the effective dimensionality of LLMs activations is very low [2].

Another perspective is that the random matrices in our method are just a convenient and efficient way to span part of the hyperspace of possible LoRA updates. Theory-wise, our method works the same way as LoRA, but we obtain more flexibility with our 'mixture-of-random-vectors' setup. To this end, we've conducted a new analysis on the expressivity of LoRA vs ELoRA on the task of fitting random square matrices.
We find that ELoRA performs as well as LoRA in this task for given number of trainable parameters, yet at the same time giving more flexibility, e.g. by allowing going to much lower parametrisations (below LoRA's rank=1). The resulting figure and analysis  will be added in the new appendix. Thank you for this fruitful suggestion!


### Experiments on vision models (JAXk, rHtj)

We conducted additional experiments by adapting Visual Transformers (pretrained on ImageNet-21k), base and large variants, on  image classification tasks of CIFAR100, Food101, Flowers102, RESISC45.
For each dataset we train on a subset of 10 samples per class, and evaluate on the full test set (CIFAR100, Food101, Flowers102) or on all the remaining samples (for RESISC45, as it does not have a test set).

We evaluated LoRA and ELoRA methods applied on the query and value layers of ViT, along with two baselines - fully-finetuned model (*Full*), and training the classification head only (*Head*). Similarly to the GLUE benchmark, we use rank 8 for LoRA, and rank 256 for ELoRA (i.e. ELoRA uses <10% of LoRA's parameters). We conducted grid-search of learning rates for all methods and reported results after 10 epochs. The reported parameter count excludes the classification head, which has to be additionally trained in all methods.

**Table 1 Finetuning ViTs on image classification datasets**

|Method | Parameters | CIFAR100 | Food101 | Flowers102 | RESISC45 |
|-------|-----------:|---------:|--------:|--------:|---------:|
| **ViT-B**
| +Head  |            | 77.7    | 86.1   |98.4| 67.2    |
| +Full  | 85.8M      |**86.5** |**90.8**|98.9|**78.9** |
| +LoRA  | 294.9K     | 85.9    | 89.9   |98.8| 77.7    |
| +ELoRA | **24.6K**  | 84.8    | 89.0   |**99.0**| 77.0    |
||||
|**ViT-L**|
| +Head  |            | 79.4    | 76.5   |98.9| 67.8    |
| +Full  | 303.3M     | 86.8    | 78.7   |98.8|**79.0** |
| +LoRA  | 786.4K     | 87.0    |**79.5** |99.1| 78.3    |
| +ELoRA |**61.4K**   |**87.5** | 79.2   |**99.2**| 78.6    |


From Table 1, we find that ELoRA approaches performance of LoRA on the *Base* model for three datasets (84.8 vs 85.9, 89.0 vs 89.9, 77.0 vs 77.7) and outperforms it for Flowers102 (99.0 vs 98.8), despite using over 10x less trainable parameters. For ViT-*Large*, we even **outperform** LoRA for three datasets: CIFAR100 (87.5 vs 87.0), Flowers102 (99.2 vs 99.1) and RESISC45 (78.6 vs 78.3). Notably, for ViT-L, ELoRA trains only a tiny fraction of 0.002% of the backbone's number of parameters.
These results overall confirm our findings in the NLP domain that ELoRA can match LoRA performance while being far more parameter-efficient. We will add these results to the paper.

**References**
 - [1] Lindenstrauss et al. Extensions of Lipschitz mappings into a Hilbert space. 1984
 - [2] Aghajanyan et al. Intrinsic Dimensionality Explains the Effectiveness of Language Model Fine-Tuning. ACL 2021

---

### Author Response · Authors · 2023-11-22
**Revised version of the paper**

We thank all the reviewers for their valuable feedback and insightful comments. In response to the suggestions received, we have uploaded a revised version of the paper. For clarity and convenience, we summarize the key changes below:
- Cited Lightweight DreamBooth in "Related Work" section under "Random Models and Projections", see page 2
- Added subsection on E2E experiments, see pages 5 and 6
- Added subsection on image classification tasks, see pages 7 and 8
- Updated subsection on instruction tuning - moved previous results and samples to appendix, added results on MT-Bench, see pages 6 and 7
- Moved paragraph "Relative performance gain" with corresponding plot from subsection on GLUE benchmark to appendix, see page 16
- Added section in the appendix on the impact on training time and GPU memory usage, see page 16
- Added section in the appendix on similarities of trained ELoRA and LoRA weights, see pages 16 and 17
- Added section in the appendix on the expressivity of ELoRA, see page 17
- Updated results for RoBERTa-Base on GLUE benchmark with data for full sequence length (512), instead of 128, to match the setup used in LoRA paper, see page 6
- Updated abstract to incorporate information about new experiments and make it more concise
- Updated "Conclusion" section to incorporate information about new experiments on image classification tasks, see page 9

---

### Meta-Review · Area_Chair_dLfe · 2023-12-04

**Metareview:**

This paper proposes to improve the efficiency of LoRA by using the same shared (randomly initialized) matrices for each layer, and then only learning scaling factors. The main strength of this approach is its simplicity, and also its empirical effectiveness across vision and language models.

While the scores were quite positive (and hence I am recommending acceptance), after having carefully read through the paper, I do think the paper would benefit from further studies, including experiments with finetuning more layers LoRA finetuning across all layers often greatly outperforms just finetuning a subset of the layers), on more realistic benchmarks (E2E is not really a canonical generation task), and larger models.

**Justification For Why Not Higher Score:**

Some of the benchmarks are not super standard.

**Justification For Why Not Lower Score:**

This is a simple method that reduces the memory footprint of LoRA by a significant margin.

---

### Decision · Program_Chairs · 2024-01-16

Accept (poster)